# When Imbalance Meets Imbalance: Structure-driven Learning for Imbalanced Graph Classification

## ABSTRACT

Graph Neural Networks (GNNs) can learn representative graph-level features to achieve efficient graph classification. But GNNs usually assume an environment where both class and structure distribution are balanced. Although previous works have considered the graph classification problem under the scenario of class imbalance or structure imbalance, they habitually ignored the obvious fact that class imbalance and structural imbalance are often intertwined in the real world. In this paper, we propose a carefully designed structure-driven learning framework called ImbGNN to address the potential intertwined class imbalance and structural imbalance in graph classification. Specifically, we find that feature-oriented augmentation (e.g., feature masking) and structure-oriented augmentation (e.g., edge perturbation) will have differential impacts when applied to different graphs. Therefore, we design optional augmentation based on the average degree distribution to alleviate structural imbalance. Furthermore, based on the imbalance of graph size distribution, we utilize a similarity-friendly graph random walk to extract a core subgraph to improve the accuracy of graph kernel similarity calculation, and then construct a more reasonable kernel-based graph of graphs, thereby alleviating the class imbalance and size imbalance. Extensive experiments on multiple benchmark datasets demonstrate that our proposed ImbGNN framework outperforms previous baselines on imbalanced graph classification tasks. The code of ImbGNN is available in https://anonymous.4open.science/r/ImbGNN-E2F0.

## CCS CONCEPTS

• **Computing methodologies → Knowledge representation and reasoning**.

## KEYWORDS

Graph classification, class imbalance, structural imbalance, augmentation, graph of graphs

**ACM Reference Format:**
Anonymous Author(s). 2023. When Imbalance Meets Imbalance: Structure-driven Learning for Imbalanced Graph Classification. In *Proceedings of ACM Conference (Conference'17)*. ACM, New York, NY, USA, 9 pages. https://doi.org/10.1145/nnnnnnn.nnnnnnn

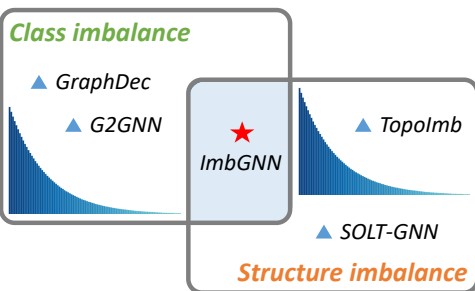

**Figure 1: Motivation of ImbGNN. The imbalance in graphs can usually be divided into category imbalance and structural imbalance. Existing methods typically only support a single imbalance environment and cannot cope with intertwined imbalances. ImbGNN, on the other hand, uses a structure-driven learning framework to simultaneously deal with multiple imbalances.**

## 1 INTRODUCTION

Due to the Zipfian [41] distribution of samples in nature, imbalanced data is prevalent in various fields, including network data [7, 12, 20, 34, 38]. For example, online discussion topics on Reddit are diverse, and the distribution of topics is naturally imbalanced due to differences in popularity. However, most datasets involved in deep graph classification tasks are artificially collected. To simplify the task, they usually assume that the data is uniformly distributed, that is, different classes have similar amounts of data. This assumption does not align with reality [27]. Therefore, graph neural networks trained on artificially collected graph datasets often fail to exhibit robust performance when directly applied to real-world applications such as social network analysis.

In the past few decades, researchers have conducted a lot of impactful work on class imbalance, especially for graph data [17, 21, 22, 29]. Typically, GraphSMOTE [40] proposed a technique inspired by SMOTE [3], generating new node representations by averaging two sampled minority class nodes. Inspired by Mixup [37], some mixed node synthesis work has also promoted the development of the field, such as GraphMixup [31], GraphENS [19], and Graph-SANN [14]. However, most of the work in this field focuses on node-level classification tasks, and it is difficult to robustly migrate to graph-level classification due to the lack of consideration for overall structural information. Recently, some work has begun to address the challenges at the graph level. For example, G$^2$GNN [30] solves the class imbalance problem by constructing abstract high-level graphs, while SOLT-GNN [16] starts from the structural level and migrates the knowledge of the structural-head graph to the structural-tail graph to alleviate structural imbalance. However,

*existing studies have focused only on a single relatively ideal imbalanced environment, which cannot simulate the data distribution in the real world.*

**Challenges.** Compared to the visual field, the graph imbalance problem in the real world is usually intertwined. Focusing only on graph class imbalance may ignore the differences in intra-class topology, leading to distortion of structural information. Focusing only on graph structural imbalance ignores the objective fact that classes usually follow a long-tailed distribution, leading to model bias towards specific classes. In addition, existing imbalanced graph classification methods usually adopt a one-size-fits-all augmentation strategy, ignoring the compatibility between the sample topology and the augmentation method, thus producing negative effects. However, there is currently no work that simultaneously focuses on these intertwined imbalances, as well as the rationality of augmentation strategies in imbalanced graph learning. Therefore, two unresolved challenges need to be addressed:

- How to mitigate the negative impact of rigid augmentation strategies on imbalanced graph learning?
- How to deal with the complex intertwined imbalances in graph classification problems simultaneously?

With these challenges in mind, we propose a carefully designed structure-driven learning framework called ImbGNN, which can simultaneously address the intertwined class imbalance and structural imbalance problems. Specifically, for the first challenge, we find that feature-oriented augmentation and structure-oriented augmentation have different effects on different types of graphs. Therefore, we design a degree-oriented optional augmentation, which dynamically adjusts the probability distribution of augmentation according to the average degree of the graph. This not only enhances sample diversity but also avoids the loss of original information, thereby alleviating structural imbalance at the degree level. For the second challenge, in addition to the flexible augmentation module, we also improve the existing graph construction method and propose a size-oriented graph of graphs construction. It achieves more accurate and reasonable connections through similarity-friendly graph random walk, thereby alleviating structural imbalance at the size level. For the constructed high-level graphs, we further use GoG propagation to allow tail-class graphs to obtain sufficient information from adjacent graphs as much as possible, thereby alleviating class imbalance. We conduct extensive experiments on five benchmark datasets including social networks and prove the superior performance of ImbGNN in imbalanced graph classification tasks.

Our **contributions** in this paper are summarized as follows:

- *New insight and framework*: for the first time, we propose a structure-driven learning framework called ImbGNN to simultaneously address the potential intertwined class imbalance and structure imbalance in graph classification.
- *New advisable augmentation*: we propose a degree-oriented optional augmentation to adapt to the graph degree imbalance problem, which can increase graph diversity while minimizing damage to original information.
- *New graph construction*: we propose a size-oriented graph of graphs construction, which uses the similarity-friendly graph random walk and GoG propagation to alleviate class imbalance

and graph size imbalance. This allows graphs to obtain rich information, resulting in high-quality representations.
- *Compelling empirical results*: ImbGNN achieves the SOTA performance across various graph benchmark datasets.

## 2 RELATED WORK

**Imbalanced Graph Classification:** Imbalanced graph classification is a challenging problem in the field of graph neural networks (GNNs). Like imbalanced node classification [6, 31, 36, 40], it commonly arises in real-world scenarios (e.g., imbalanced social network classification) where class distributions of labeled graphs are skewed [15]. Several methods have been proposed to address this issue, such as Graph-of-Graph Neural Networks ($G^2$GNN), which derive extra supervision globally from neighboring graphs and locally from stochastic augmentations of graphs [30]. In addition to class imbalance, graph-level structural imbalance, such as graph size imbalance, has also received attention. Typically, SOLT-GNN first identifies co-occurrence patterns in the structures of larger, or "head", graphs, to generate transferable knowledge for smaller, or "tail", graphs [16]. Although progress has been made, this task has not been studied in depth relative to imbalanced node classification task. Existing studies have focused only on a single relatively ideal imbalanced environment, which cannot simulate the data distribution in the real world. When one imbalance meets another imbalance, there is still a lack of a reliable solution to deal with the intertwined imbalance problem. Fortunately, *our proposed ImbGNN systematically alleviates multiple graph-level imbalances with one framework and is more suitable for imbalanced graph learning in open environments.*

**Graph Data Augmentation:** Graph Data Augmentation (GraphDA) has been widely used in many fields because it can effectively alleviate overfitting and improve model generalization performance [8]. GraphDA is simple to design and can be implemented through graph processing to achieve various DA, such as masking node and dropping edge. Recently, G-mixup proposed a mixed-based GraphDA method, which improves the robustness of the model by fusing two graphs and their labels [11]. In the computer vision domain, some studies have tried to randomly or sequentially combine DA, such as AutoAugment [4], Fast AutoAugment [13], and RandAugment [5]. However, the augmentation methods in the graph domain still lack flexibility and ignore the potential correlation between DA and the distribution of graph classes and structures. Therefore, it may cause the augmented graph to lose critical topological information in an imbalanced environment. Therefore, *it is necessary to design an imbalanced GraphDA that can improve diversity while minimizing damage to original information.*

**Graph of Graphs:** Graph of Graphs (GoG) is a graph-based model that can learn from multiple graphs. It is a generalization of the GNN model [18]. The key idea behind GoG is to represent each graph as a node in a higher-level graph, which is called the meta-graph. The edges in the meta-graph represent the relationships between the graphs. Recently, [10] and [26] leverage GoG to solve link prediction and graph classification. To break the limitation of providing GOG in advance, [30] construct a kNN GoG based on graph topological similarity and aggregate neighboring graph

information by propagation on the constructed GoG to solve imbalanced graph classification. However, this fair connection method ignores the potential problem of structural imbalance and cannot give more attention to the structural-tail graph. Similar to handling class imbalance, *structural-tail graphs need to integrate more information from multiple graphs to improve the classification accuracy of the model.*

## 3 IMBGNN: A STRUCTURE-DRIVEN GNN LEARNING FRAMEWORK

In this section, we introduce our proposed structure-driven ImbGNN framework. We illustrate the overall framework in Figure 2. We analyze and alleviate the imbalance problems existing in graph classification from three aspects: *class imbalance*, *graph's average degree imbalance*, and *graph size imbalance*. First, we design degree-oriented optional augmentation to adapt to the graph degree imbalance problem, which samples augmentation methods from different distributions for degree-head and degree-tail graphs. Next, we proposed to construct graphs of graphs (GoG) based on the graph size imbalance and perform a similarity-friendly graph random walk on large graphs for subgraph sampling to improve the accuracy of graph similarity calculation. The information propagation of GoG also enables tail classes to share some information, thereby improving the model's discrimination in tail classes. In addition, we also design a size-based GoG connection method to alleviate the size imbalance. Based on the above structure-driven framework, ImbGNN can comprehensively cope with the scenario of imbalance meets imbalance.

### 3.1 Preliminary

**Problem Formulation:** A graph can be expressed as $G = \{V, E, X\}$, where $V$ is the node set, $E$ is the edge set, $X \in \mathbb{R}^{|V| \times d}$ is the initial feature matrix of the node, and $d$ is the dimension of the feature. In addition, we denote the neighbor set of node $u$ in the graph as $N_u$.

Given a graph set $\mathcal{G} = \{G_1, G_2, ..., G_N\}$, where $G_i = \{V_i, E_i, X_i\}$, and their corresponding label sets $Y = \{y_1, y_2, ..., y_N\}$, the goal of graph-level representation learning is to learn a mapping function $\mathcal{F} : G \rightarrow \mathbb{R}^f$ to map the graph to a low-dimensional vector $h_{G_i} \in \mathbb{R}^f$. This low-dimensional vector is then fed to a classifier to obtain the predicted label distribution, thereby obtaining the predicted output of the sample.

**Graph Neural Networks:** Graph neural networks (GNNs) are a type of deep learning model that operates on graphs. These models typically rely on the key operation of neighborhood aggregation, recursively passing and transforming messages from neighboring nodes to form the representation of the target node. This process can be represented as:

$$\mathbf{h}_v^l = \text{AGGREGAT}\left(\mathbf{h}_v^{l-1}, \left\{\mathbf{h}_i^{l-1} : i \in N_v\right\}; \theta_g^l\right), \quad (1)$$

where $h_v^l$ represents the feature representation of node $v$ in the $l$-th GNN layer. $\text{AGGREGAT}(\cdot; \theta_g^l)$ represents the neighborhood aggregation function of the $l$-th layer with $\theta_g^l$ as its parameter. Note that $h_v^0$ is initialized by $X_v$. We perform GNN propagation for $L$ times to obtain the output representation of nodes. The representation of the entire graph $h_G$ is obtained by combining the output

representations of all nodes using a READOUT function.

$$h_G = READOUT(\{h_v : v \in V\}) \quad (2)$$

where the READOUT function is usually permutation invariant, such as summation, averaging, etc.

### 3.2 Degree-oriented Optional Augmentation

In order to improve the generalization performance of deep models, researchers usually perform various data augmentations on training data to obtain more representative samples. For graph data, data augmentation methods can usually be divided into structure-oriented augmentation and feature-oriented augmentation. Considering that structural information and feature information are equally important to graph features, we choose the two most commonly used augmentations: dropping edge and masking node.

**Dropping edge:** We perform some perturbations on the given graph structure by randomly dropping edges while keeping node order and features unchanged. Here, we randomly set part 1 to 0 in the adjacency matrix, which can be defined as follows:

$$\tilde{A} = A \wedge C \quad (3)$$

where $A$ is the adjacency matrix of the input graph, $C$ is the dropping matrix, and $\wedge$ represents the AND operation. Dropping matrix $C$ is obtained by sampling, i.i.d., from a prior distribution, and $C_{ij} = 1$ means keeping the original existing edges, $C_{ij} = 0$ means discarding the original existing edges. For example, assuming a dropping ratio $\rho$ is given, we can define the dropping matrix $C$ as $C_{ij} \sim \text{Bernoulli}(\rho)$, that is, the elements in $C$ have a probability of $\rho$ to be set to 1 and a probability of $1 - \rho$ to be set to 0.

**Masking node:** We do not directly delete nodes that may disconnect the original graph into several non-connected blocks. Instead, we set a part of the entries in the node feature matrix $X$ to 0, that is, we do not change the topological structure information of the graph, and only perform node mask in the feature dimension. It can be defined as follows:

$$\tilde{X} = X \odot M \quad (4)$$

among them, $X \in \mathbb{R}^{n \times d}$ represents the original feature matrix of the graph, $M$ is a masking vector, $M_i = 0$ represents that the feature of node $i$ is masked, i.e., $\tilde{X}_i = 0$, otherwise $M_i = 1$ means node $i$ does not change and maintains the status quo, i.e., $\tilde{X}_i = X_i$. Finally, the masked feature matrix $\tilde{X} \in \mathbb{R}^{n \times d}$ is obtained.

Existing methods usually treat each sample equally when using graph augmentations. However, an indisputable fact is often easily overlooked, i.e., *samples are different in terms of structure and features, and forcing certain augmentations is unfair and will inevitably cause information loss.* As mentioned in CUDA [1], we hope to generate hard samples through data augmentation to improve the generalization ability of the model, and we also hope that data augmentation will lose as little of the original information of the samples as possible. As shown in Figure 1, the graph's average degree follows a long-tailed distribution. We believe that for those graphs with a large average degree (degree-head graph), the structural information is sufficient, and dropping edges will not have a drastic impact on the original information of the graph. For degree-tail graphs, the number of edges is scarce. Adopting

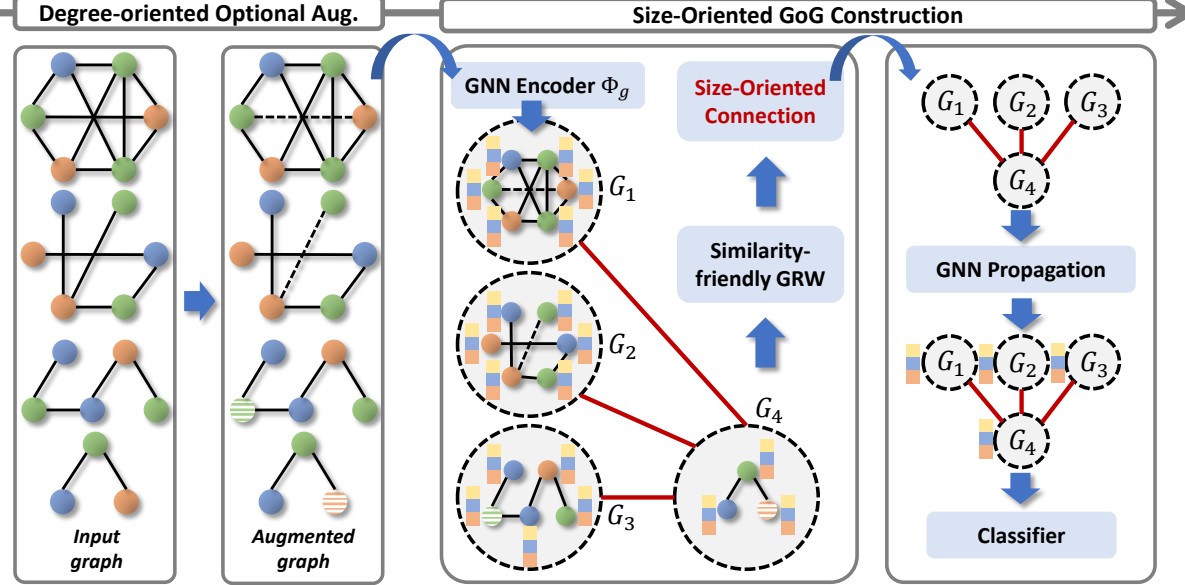

**Figure 2: Overall of our proposed structure-driven ImbGNN framework. Degree-oriented optional augmentation is designed to solve the potential bias of augmentation methods for graphs with different average degrees. The graphs obtain independent graph representation through a GNN Encoder. During the preprocessing of graph similarity, we designed a method called size-oriented GoG construction, which constructs GoG edges unfairly for different sizes. Finally, information is propagated in the high-level graph, and the classification result is obtained through a classifier.**

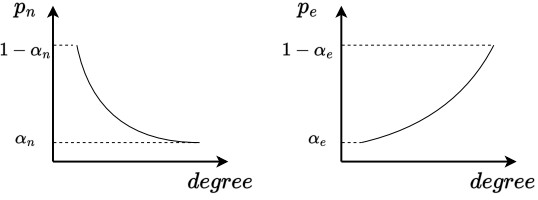

**Figure 3: The trend of the function $p_e$ and $p_n$ as it varies with the average degree of the graphs.**

a structure-oriented augmentation method like dropping edges is likely to lose the original information of the sample, leading to abnormal topological information. This type of graph is more suitable for a feature-oriented augmentation method like masking nodes. We also confirmed our guess through experiments. Based on the above ideas, as shown in Figure 4(a), we design a degree-oriented optional augmentation (DoOA). For each sample, we respectively sample the augmentation probability coefficients $q_n$ and $q_e$ corresponding to node masking and edge dropping and set two augmented thresholds as $\alpha_n$ and $\alpha_e$. Formally, we define these two degree-oriented augmentations as:

$$O_e(G_i) = \begin{cases} Aug_e(G_i), & \text{if } q_e(i) \le p_e(deg(G_i)) \\ G_i, & \text{otherwise} \end{cases}$$

$$\text{where } p_e(deg) = (\alpha_e - (1 - \alpha_e)) * \frac{\ln\left(\frac{deg}{D_0}\right)}{\ln\left(\frac{D_1}{D_0}\right)} + 1 - \alpha_e \quad (5)$$

and

$$O_n(G_i) = \begin{cases} Aug_n(G_i), & \text{if } q_n(i) \le p_n(deg(G_i)) \\ G_i, & \text{otherwise} \end{cases}$$

$$\text{where } p_n(deg) = ((1 - \alpha_n) - \alpha_n) * \frac{\ln\left(\frac{deg}{D_0}\right)}{\ln\left(\frac{D_1}{D_0}\right)} + 1 - \alpha_n \quad (6)$$

so the composite degree-oriented optional augmentation can be defined as:

$$O(G_i; p_n, p_e) = O_e \circ O_n(G_i) \quad (7)$$

where $\alpha_n$ and $\alpha_e$ are two hyperparameters, $D_0 = MIN(\overline{D}(G_i))$ and $D_1 = MAX(\overline{D}(G_i))$. For any $i$, $\overline{D}(G_i) = \sum_{j=1}^{|V_i|} degree(v_j)$ is the average degree of all nodes in $G_i$. For any graph $G_i$, $q_n(i)$ and $q_e(i)$ are two values randomly sampled at $[0, 1]$. In addition, whether to use a dropping depends on the relationship between $q_e(i)$ and the threshold $p_e(deg(G_i))$, i.e., if $q_e(i) \le p_e(deg(G_i))$, then $G_i$ will be augmented with a dropping edge, otherwise it will not be used. As the average degree of graph increases, the threshold $p_e(deg)$ gradually increases, and the probability of using dropping edges gradually increases. For the masking node, the trend of the threshold $p_n(deg)$ is exactly the opposite. As the average degree of the graph decreases, the threshold gradually increases, and the probability of using the masking node gradually increases. The distribution that the threshold $p_e(deg)$ and $p_n(deg)$ obeys is shown in Figure 3. The final augmented graph is the result of two composite operations $O_n$ and $O_e$. Through degree-oriented optional augmentation, we can adaptively apply appropriate augmentation

methods, drop-dominated or mask-dominated, to each input graph, thereby alleviating the structural imbalance at the degree level.

### 3.3 Size-oriented GoG Construction

For the augmented graph, we further analyze and resolve imbalances at other levels, such as class or graph size. When dealing with traditional class imbalance problems, the most well-known method SMOTE enriches the information of the minority class through feature interpolation. As G$^2$GNN [30] does, we draw on feature propagation and aggregation mechanisms like SMOTE [3] and Mixup [37] to construct graphs of graphs (GoG). Specifically, we regard each independent input graph as a node, and the features of the input graph after being processed by the GNN encoder as the features of this node. We connect and reconstruct these nodes so that feature information can propagate between different graphs, thereby enabling minority classes to obtain information from other samples to enrich their own features. The connection of GoG is based on the similarity of graphs, and we connect graphs with higher similarity. The calculation of similarity is done before training and is a one-time operation. Inspired by SOLTGNN [16], graph size usually also exhibits a long-tailed distribution, and this feature has certain problems when constructing GoG: (1) The difference in graph size will lead to a decrease in accuracy when calculating graph similarity. As shown in Figure 4(b), a large graph and a small graph do not have high similarity under the calculation of the shortest path kernel. However, there is a subgraph in the large graph that has extremely high similarity with the small graph. Therefore, we should mine more fine-grained local information. (2) Graphs with a smaller size often lack structural information and have a low upper limit on their own information, which can lead to biased classification results. Therefore, we design a size-oriented graph of graphs (SoGoG) construction method.

For a given graph set $\mathcal{G}$, we construct an abstract high-level graph $\mathbb{G} = (\mathbb{V}, \mathbb{E})$, where $G_i \in \mathcal{G}$, corresponding to a node $\mathbb{V}_i$ in $\mathbb{G}$. We pass the augmented graph $\tilde{G}_i$ processed by the DoOA module as input into the GNN Encoder $\phi_g(\tilde{G}_i; \theta_g)$ to obtain the features of all nodes. Finally, the READOUT function aggregates all node features into the feature $H_i$ of the graph $G_i$. So the initial node feature corresponding to $\mathbb{V}_i$ in the high-level graph is $H_i$. If $G_i$ and $G_j$ are similar enough, then $\mathbb{V}_i$ and $\mathbb{V}_j$ will be connected. We believe that two graphs with sufficiently similar topological information are highly likely to belong to the same class. Therefore, through the information propagation of the high-level graph $\mathbb{G}$, we can make the feature information propagate as much as possible within the same class, thereby alleviating the defect of insufficient information in minor class samples.

**Similarity-friendly Graph Random Walk:** To achieve accurate connections, we use the shortest path kernel to calculate the similarity between graphs, resulting in $simi_{pre}(i, j) = \Omega(G_i, G_j)$. However, as mentioned earlier, we need to consider the impact of graph size imbalance on similarity calculation. For this, we propose a similarity-friendly graph random walk (GRW), which constructs core subgraphs to achieve more accurate similarity calculation. Specifically, for the imbalance ratio $\rho_s$, we define the graph of $|G_i| \geq K$ as the size-head graph, and the graph of $|G_i| < K$ as the size-tail graph, where $|G|$ represents the number of nodes in

graph $G$. We perform the GRW operator on all size-head graphs to reduce the size of the size-head graphs, in the hope of achieving more accurate similarity calculation between them and the size-tail graphs, as shown in Figure 4(b). Since it is difficult for GRW to accurately control the final subgraph size, we expect that the obtained subgraph size is within the interval $[0.9K, 1.1K]$. We further calculate the similarity $simi_{post}(i, j) = \Omega(G'_i, G'_j)$, where

$$G'_i = \begin{cases} G_i, & \text{if } G_i \text{ is size-tail graph} \\ RW(G_i), & \text{if } G_i \text{ is size-head graph} \end{cases} \quad (8)$$

$RW(G_i)$ is the subgraph obtained by performing a GRW on $G_i$. The final similarity is $simi(i, j) = Max(simi_{pre}(i, j), simi_{post}(i, j))$.

The traditional random walk operator is prone to passing through nodes that have already been traversed and falling into local loops, which slows the growth of the subgraph size and reduces the efficiency of preprocessing graph similarity. Therefore, we draw on CNARW [28], a fast random walk algorithm with common neighbor awareness, to alleviate the problem of slow convergence speed to $[0.9K, 1.1K]$ when performing random walks in large-scale graphs. In traditional RW, for the current node $u$, the probability of any node $v \in N(u)$ becoming its next hop node is $\frac{1}{deg(u)}$, where $N(u)$ represents the neighbor set of node $u$, and $deg(u)$ represents the node degree of $u$. And CNARW is a kind of weighted walking. For any node $v \in N(u)$, if node $v$ has a higher degree or fewer common neighbors with node $u$, then the probability of $v$ being sampled as the next hop node of $u$ is higher. This weighted approach can more easily explore nodes that have not been visited before. Because if a node has a higher degree, the number of unknown nodes that can be reached by exploring from it is greater. And if $v$ and $u$ have fewer common neighbors, the probability of walking back to the previously explored node from $v$ will be lower. Therefore, CNARW has a higher probability of sampling nodes that have not been visited before, and it reduces the probability of resampling nodes that have been sampled in future random walk samplings. Formally, the node sampling probability matrix $P$ is expressed as follows:

$$P_{uv} = \begin{cases} \tilde{p}_{uv}/(1 - \tilde{p}_{uu}), & \text{if } v \in N(u) \\ 0, & \text{otherwise} \end{cases} \quad (9)$$

where the calculation of $\tilde{p}_{uv}$ is:

$$\tilde{p}_{uv} = \begin{cases} \frac{1}{\deg(u)} \times \left(1 - \frac{C_{uv}}{\min\{\deg(u), \deg(v)\}}\right), & \text{if } v \in N(u) \\ 1 - \sum_{k \in N(u)} \tilde{p}_{uk}, & \text{if } v = u \\ 0, & \text{otherwise} \end{cases} \quad (10)$$

where $C_{uv}$ represents the number of common neighbors of node $u$ and node $v$.

**Size-oriented Connection:** Based on the calculated similarity matrix $simi$, we can construct edges on the high-level graph $\mathbb{G}$. Previous construction methods usually adopt a fair edge connection method, that is, connecting each graph $G_i$ to its $k$ graphs with the highest similarity to construct a kNN graph. Considering the long-tailed distribution of graph size, we believe that size-tail graphs need to integrate more information from graphs of the same class to improve their classification accuracy. Therefore, we propose an unfair edge construction method based on graph size. The graph $G_i$ will connect edges with $S_i$ graphs with the highest similarity.

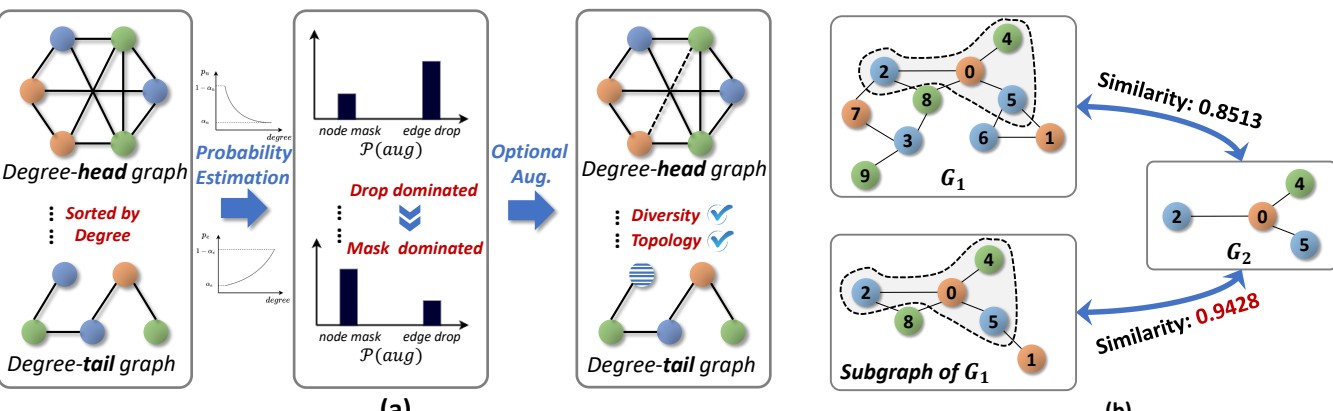

Figure 4: (a) The overall framework of degree-oriented optional augmentation (DoOA). (b) Sample subgraphs has the potential to improve graph similarity calculations.

For size-tail graph, $S_i = k_t$ and for size-head graph, $S_i = k_h$, which satisfies $k_t > k_h$. We use this unfair size-oriented connection to alleviate the problem of the model's poor classification performance for size-tail graphs under the condition of graph size imbalance. The GoG constructed by the above method has an adjacency matrix $\mathbb{A}$, which is expressed as follows:

$$\mathbb{A}_{ij} = \begin{cases} 1, & \text{if } i \in \mathbb{N}_j \text{ or } j \in \mathbb{N}_i \\ 0, & \text{otherwise} \end{cases} \quad (11)$$
$$\text{where } \mathbb{N}_i = \text{argMAX}(simi, i, S_i)$$

where argMAX$(simi, i, S_i)$ represents the indices of the $S_i$ largest values in the similarity vector corresponding to node $\mathbb{V}_i$ in $simi \in \mathbb{R}^{|\mathbb{V}| \times |\mathbb{V}|}$.

**Graph of Graphs (GoG) Propagation:** After constructing the high-level graph, the propagation process of GoG can be expressed as:

$$\mathbf{P}^{l+1} = \mathbb{D}^{-1}\mathbb{A}\mathbf{P}^l, l \in \{1, 2, \dots, L\} \quad (12)$$

where $\mathbb{D}$ is the degree matrix, $\mathbf{P}^0 = \mathbf{H}$ is the feature vector $H_i$ of all individual graphs $G_i$ which are previously obtained from GIN followed by the graph pooling matrix. Finally, the distribution of predicted labels is obtained through the linear layer, as shown below:

$$\mathbf{B} = \text{MLP}(\mathbf{P}^L) \quad (13)$$

Through the information propagation of GoG, the information between samples of the same class has been fully interacted and complemented. For tail classes, their intra-class information has been supplemented, which narrows the information gap with head classes and thus improves the class imbalance problem.

## 4 EXPERIMENT

In this section, we conduct extensive experiments of imbalanced graph classification on various graph datasets with different levels of imbalance to evaluate the effectiveness of ImbGNN, and further carry out adequate ablation experiments to provide a better perspective to perceive the superiority of ImbGNN to solve the alleviate imbalance problem.

Table 1: Statistics of datasets (# denotes the "number").

| Dataset | # Graphs | # Avg-Node | # Avg-Edge | # Attr |
|---|---|---|---|---|
| PTC-MR [24] | 344 | 14.29 | 14.69 | 18 |
| NCI1 [25] | 4110 | 29.87 | 32.30 | 37 |
| PROTEINS [2] | 1113 | 39.06 | 72.82 | 3 |
| D&D [9] | 1178 | 284.32 | 715.66 | 89 |
| REDDIT-B [33] | 2000 | 429.63 | 497.75 | / |

## 4.1 Setup

**Datasets:** We utilize a total of five benchmark datasets in our study. The Reddit-B [33] dataset represents social networks, while the D&D [9] and PROTEINS [2] datasets are from the field of bioinformatics. The NCI1 [25] and PTC-MR [24] datasets represent chemical compounds. The statistics of these datasets are summarized in Table 1.

**Baselines:** To assess the efficacy of our model, we compare it with various rebalancing methods, including vanilla, up-sampling, and re-weighting. The vanilla method does not involve any rebalancing operation during the training process. Up-sampling, a traditional approach to addressing imbalance issues, involves repeating samples from the minority class to achieve class balance. Re-weighting is a cost-sensitive method that applies different weights to different classes when calculating loss. Following the approach in [39], we set the weight of each class to be inversely proportional to the number of training samples it contains, thereby assigning higher weights to minority classes. For each rebalancing method, we run three baseline methods: GIN [32], InfoGraph [23], and GraphCL [35]. Additionally, we also evaluate two versions of G$^2$GNN [30] (i.e., remove-edge and mask-node) for effective comparison.

**Evaluation Metrics:** To more accurately evaluate the performance of our model, we adopt two metrics commonly used in previous imbalanced classification work: F1-Macro and F1-Micro. F1-Macro computes the accuracy for each class and then averages these values to yield the final result, treating different classes equally, akin to

**Table 2: Imbalanced class graph classification results on five benchmark datasets. The numbers following each dataset name represent the imbalance ratios between minority and majority class. Blod indicates the best performance while underline indicates the second best.**

| Method | Base model | PROTEINS (30:270) | | D&D (30:270) | | NCI1 (100:900) | | PTC-MR (9:81) | | REDDIT-B (50:450) | |
|---|---|---|---|---|---|---|---|---|---|---|---|
| | | F1-macro | F1-micro | F1-macro | F1-micro | F1-macro | F1-micro | F1-macro | F1-micro | F1-macro | F1-micro |
| Vanilla | GIN [32] | 25.33 | 28.50 | 9.99 | 11.88 | 18.24 | 18.94 | 17.74 | 20.30 | 33.19 | 36.02 |
| | InfoGraph [23] | 35.91 | 36.81 | 21.41 | 27.68 | 33.09 | 34.03 | 25.85 | 26.71 | 57.67 | 67.10 |
| | GraphCL [35] | 40.86 | 41.24 | 21.02 | 26.80 | 31.02 | 31.62 | 24.22 | 25.16 | 53.40 | 62.19 |
| Up-sampling | GIN [32] | 65.64 | 71.55 | 41.15 | 70.56 | 59.19 | 71.80 | 44.78 | 55.43 | 66.71 | 83.00 |
| | InfoGraph [23] | 62.68 | 66.02 | 41.55 | 71.34 | 53.38 | 62.20 | 44.29 | 48.91 | 67.01 | 78.68 |
| | GraphCL [35] | 64.21 | 65.76 | 38.96 | 64.23 | 49.92 | 58.29 | 45.12 | 53.50 | 62.01 | 75.84 |
| Re-weighting | GIN [32] | 54.54 | 55.77 | 28.49 | 40.79 | 36.84 | 39.19 | 36.96 | 43.09 | 45.17 | 51.92 |
| | InfoGraph [23] | 65.73 | 69.60 | 41.92 | 72.43 | 53.05 | 62.45 | 44.09 | 49.17 | 65.79 | 77.35 |
| | GraphCL [35] | 63.46 | 64.97 | 40.29 | 67.96 | 50.05 | 58.18 | 44.75 | 52.22 | 62.79 | 76.15 |
| $G^2$GNN [30] | Remove edge | 67.70 | 73.10 | 43.25 | 77.03 | 63.60 | 72.97 | 46.40 | 56.61 | 68.39 | 86.35 |
| | Mask node | 67.39 | 73.30 | 43.93 | 79.03 | 64.48 | **74.91** | 46.61 | 56.70 | 67.52 | 85.43 |
| ImbGNN | / | **67.90** | **73.95** | **46.39** | **83.42** | **65.52** | 74.54 | **47.86** | **60.73** | **69.01** | **86.77** |

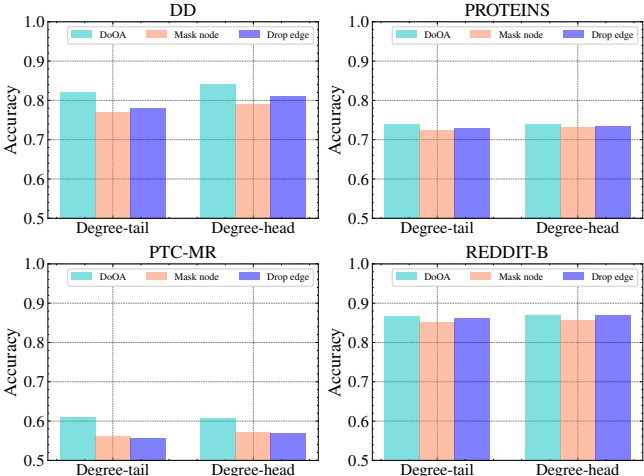

Figure 5: Ablation study on data augmentation methods, we compared the performance of our proposed DoOA with simply using dropping edge or masking node on graphs with different degree properties.

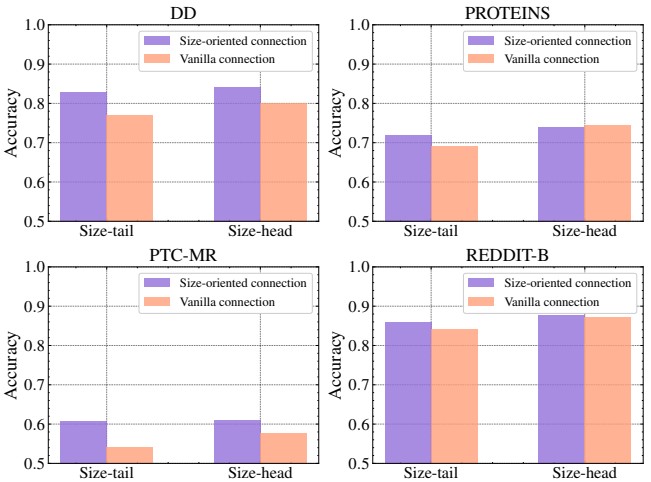

Figure 6: Ablation study on GoG connection methods, we compared the performance of our proposed size-oriented connection with vanilla connection on graphs with different size properties.

balanced accuracy (bAcc.). F1-Micro, on the other hand, calculates the accuracy across all samples, which may result in the majority class dominating the process, similar to overall accuracy (Acc.).

**Settings:** For each dataset, we categorize graphs as either size-tail graph/size-head graph or degree-tail graph/degree-head graph, with the division following the traditional handling of long-tail problems, using K as the dividing value. The choice of K can be freely determined but typically follows the Pareto principle (i.e., the 20/80 rule), selecting the top 20% of large graphs as head graphs, with the remainder as tail graphs. In unfair size-oriented connections, we choose values for $k_t$ from [2,3] and for $k_h$ from [1,2].

## 4.2 Main Results

In this section, we compare the performance of ImbGNN with the aforementioned baselines in the graph classification task. Mimicking the division of long-tail datasets in imbalanced image classification and the settings in $G^2$GNN, we select one class from the dataset as the minority class and reduce the samples of this class in the training set until the imbalance ratio reaches 1:9, thereby creating an extreme class imbalance scenario. The results are reported in Table 2.

As can be observed from Table 2, ImbGNN achieves the best results on almost all five datasets under both F1-macro and F1-micro metrics, with only a slight underperformance on F1-micro

**Table 3: Impact of connection num $k_t$ and $k_h$ on ImbGNN.**

| $k_t$ | $k_h$ | PROTEINS | D&D | NCI1 | PTC-MR | REDDIT-B |
|---|---|---|---|---|---|---|
| 2 | 1 | 73.95 | 81.56 | 73.17 | 60.73 | 86.77 |
| 3 | 2 | 72.49 | 83.42 | 72.46 | 58.46 | 85.49 |
| 3 | 1 | 72.84 | 82.37 | 74.54 | 57.91 | 86.20 |

for the NCI1 dataset compared to G$^2$GNN. It can be seen that the vanilla scheme performs worst as it does not take into account class imbalance. For the up-sampling scheme, it requires repeated sampling of the minority class, leading to additional computational costs and potential overfitting of the minority class. The re-weighting scheme performs mediocrely, even worse than up-sampling. G$^2$GNN achieves decent results through global supervision information propagation and local self-consistency regularization, but its single augmentation method of either removing edges or masking node features has limitations. In biochemistry datasets such as D&D and PTC-MR, masking node features as a data augmentation method performs better, while in social datasets like REDDIT-B, removing edges as an augmentation method performs better. It cannot use a fixed augmentation method to deal with all datasets. This problem will be more obvious when facing unknown datasets. This is where our ImbGNN has made improvements, enhancing the robustness of the model. Our ImbGNN adopts an optional augmentation method based on the average degree of graphs, improving the model's generalization ability and better adapting to all different datasets.

### 4.3 Ablation Study

We have validated the effectiveness of our method in two important aspects of dealing with structural imbalance. Firstly, for degree-oriented optional augmentation (DoOA), we compare it with the results of using only edge dropping or node masking as augmentation methods. As shown in Figure 5, using DoOA for augmentation significantly improves the results for both the degree-tail graph and the degree-head graph. DoOA can effectively match a more suitable data augmentation method for graphs with different structural properties. Secondly, as shown in Figure 6, for size-oriented GoG construction, we compared it with the fair connection method in G$^2$GNN (i.e., each node in GoG is connected to its $k$ most similar nodes). The effect of using size-oriented connection is significantly better than vanilla connection. The improvement in the size-tail graph is undoubtedly significant, which also validates the conceive we proposed before. For the imbalance in size distribution, this unfair connection method can not only allow size-tail graphs to obtain richer supervision information but also reduce the possibility of size-head graphs, which already have abundant information, being affected by noise information.

### 4.4 Parameter Sensitive

For the hyperparameters $\alpha_e$ and $\alpha_n$ used in DoOA, we set their thresholds to be equal. We conduct experiments on the sensitivity of the augmentation ratio $\alpha = \alpha_e = \alpha_n$, and the results are shown in Figure 7. The experimental results show that when $\alpha > 0.5$, the

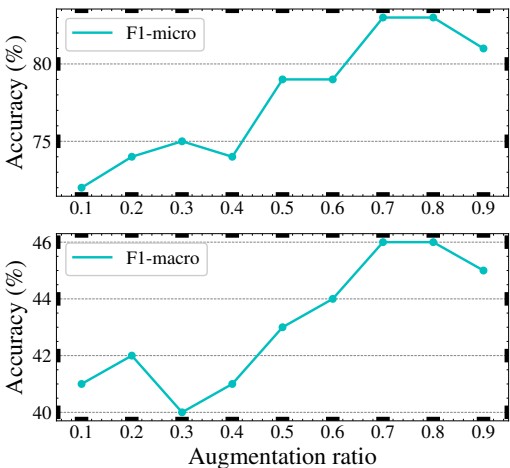

**Figure 7: Sensitivity experiments of $\alpha$ in DoOA conducted on the D&D dataset.**

performance of the model is much better than when $\alpha < 0.5$. This also confirms our previous point: for graphs with a smaller average degree, mask node feature, a feature-oriented augmentation method, is more suitable, while for graphs with a larger average degree, dropping edge, a structure-oriented augmentation method, is more appropriate. If the opposite is true, the performance will drop dramatically because for graphs with a smaller average degree, structural information is scarce to begin with. When perturbing edges, it is easy to lose key information or introduce serious noise, thereby affecting the results. Furthermore, in the Size-oriented GoG Construction, the number of connections $k_t$ and $k_h$ are hyperparameters and we also conduct a performance comparison, as shown in the Table 3. We found that the sensitivity of these two parameters is not high, and the difference is not significant when the condition $k_t > k_h$ is met.

### 5 CONCLUSION

In this paper, we focus on the problem of imbalanced graph classification from two perspectives: class imbalance and structural imbalance. The scenario widely exists in the real world, yet few studies have delved into it. To address this, we propose a novel model ImbGNN, which mitigates class imbalance while considering two aspects of structure imbalance: the average degree and size of graphs. We design a degree-oriented optional augmentation, an optional augmentation method for graphs with significant differences in average degree. Furthermore, while using GoG to address class imbalance, we design a size-oriented GoG construction method for graphs with size difference. This unfair connection method allows smaller graphs to access more information. Experiments on multiple benchmark datasets demonstrate the effectiveness of our model. In future work, we plan to explore solutions when more structure imbalances (such as topological structures) and class imbalances intertwine.

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
