# OpenReview forum: "When Imbalance Meets Imbalance: Structure-driven Learning for Imbalanced Graph Classification"
_ACM.org/TheWebConf/2024/Conference — TheWebConf24_

### Official Review · Reviewer_3wWc · 2023-10-30

**Novelty:** 4
**Technical Quality:** 4

**Review:**

This paper focuses on the fact that existing studies overlook the fact that class imbalance and structural imbalance are often intertwined. To address the structural imbalance (i.e., degree imbalance), the paper proposes a degree-oriented optional augmentation, which samples augmentation methods from different distributions for degree-head and degree-tail graphs. Moreover, to address the class imbalance, the paper devises similarity-friendly random walk, which enables tail classes to share some information, thereby improving the model’s discrimination in tail classes. Lastly, the paper proposes a size-based GoG connection method to alleviate the size imbalance.

Pros

- The paper studies an important problem, i.e., graph classification under imbalance scenarios.
- The paper is well-written and easy to follow.

Cons

- Although we can naively assume that class imbalance and structural imbalance are intertwined, it would be more nice to see how they are intertwined. For example, what impact does structural imbalance have on class imbalance and vice versa? This would elaborate on why these two imbalance situations should be considered simultaneously, rather than simply saying that none of existing studies addressed them both at the same time.
- Moreover, what is the actual negative impact of rigid augmentation strategies on imbalanced graph learning? Similar concern as the above comment.
- The meaning or semantic of “structure” in bioinformatics, chemical compounds, and social networks should be different, implying that the notion of structural imbalance should also be different among them. It would be nice to see how they are similar and different, and how they should be treated differently in the model. However, it seems that the inherent difference in nature of each dataset is overlooked while constructing the model.
- It is hard to agree with the statement that “graphs with a large average degree (degree-head graph), the structural information is sufficient, and dropping edges will not have a drastic impact on the original information of the graph.” This might be true for social networks, but for chemical compounds, a slight change in the structure (regardless of how many bonds there are) may drastically change the property of the compound itself.
- It seems that structural imbalance eventually implies degree imbalance in the method section, which is not expected as there are many other concepts in terms of structural information other than degree. In other words, the concept is introduced as if it is a broader concept, but ended up with a narrow concept.
- It is hard to agree with the statement “two graphs with sufficiently similar topological information are highly likely to belong to the same class” in Line 505. Not only topological information but also the feature information relevant to the label, and sometimes even more important indicator of the label. Moreover, this depends on the graph type, e.g., whether it is biology graph, chemical graph or social graphs.
- The concept of sample diversity in Line 148 does not seem to fit into the overall story, as it is introduced without mentioning about the diversity aspect.
- If graph augmentation is performed in each iteration, it seems that graph random walk also should be performed in each iteration, which limits the scalability of the proposed method.
- Equation 5 and 6 seem to be heuristic, which needs further explanation.
- MUTAG and DHFR are not used in the experiments without any reason even though they are used in G2GNN. However, for fair comparisons with G2GNN, they should also be considered as benchmark datasets.
- Experiments regarding various imbalance ratios would further validate the superiority of the proposed method.
- As scalability is a concern, some analysis on the scalability of the model is required.
- Inconsistent use of term:  Line 77 Category imbalance → Class imbalance
- Line 155: GoG is used without defining it.
- Line 329: n → |V|
- Line 392: It would be nice to explicitly refer to relevant experiments of the paper when mentioning “We also confirmed our guess through experiments.”

**Questions:**

Please address my questions above.

**Reviewer Confidence:**

4: The reviewer is certain that the evaluation is correct and very familiar with the relevant literature

**Scope:**

4: The work is relevant to the Web and to the track, and is of broad interest to the community

---

### Official Review · Reviewer_Zoog · 2023-11-19

**Novelty:** 4
**Technical Quality:** 5

**Review:**

This paper focus on the problem of imbalanced graph classification from both views of class imbalance and structural imbalance with a proposed ImbGNN model.
The degree-oriented optional augmentation implements average degree adaptive augmentation with edge drop and node mask, and the size-oriented GoG construction method for graphs implements unfair connections in GoG structure. Experiments on multiple
benchmark datasets, along with ablation study, parameter sensitivity analysis could verify its effectiveness to some extent.

**[Quality.]** The overall quality of this work is good, with a well-structured paper, logical methodology, and fair experimental evaluations.

**[Clarity.]** The clarity can be satisfied and relatively clear, except for better defining the structure-imbalance issue formally (node degree distribution imbalance?).

**[Originality]** Two specific adaptive techniques, average degree adaptive augmentation and unfair gog structural learning exhibit good originality and innovation.

**Questions:**

Here are some suggestions and questions:

(1) The authors claimed to jointly address the class-imbalance and structual-imbalance, whether what ever the degree-oriented augmentation or size-oriented GoG construction seems concentrate on the structural part not specific class-imbalance. More explainations should be provided here. And how the statistics of datasets class-imbalance and structual-imbalance degrees shold be provided.

(2) The definition of structural imbalance is not too clear, does it specifically mean the "node degree distribution imbalance"?

(3) Although two particular adaptive techniques appear valid, it seems that the hyperparameter threshold alphas could significantly influence the experimental outcomes.
According to Figure 7, the only 0.1 changes from 0.6 to 0.7 would lift the performance from 44 to 46. In this case, when given a dataset without knowing its balance or imbalance, how should we set such hyperparameters?

(4) Subgraph GRW with shortest path kernel seems logical to calculate different size graph's similarity, but I am worried about whether it is accurate. if a graph is similar with very small part of another large-scale graph, can we say they are really similar?

**Reviewer Confidence:**

4: The reviewer is certain that the evaluation is correct and very familiar with the relevant literature

**Scope:**

4: The work is relevant to the Web and to the track, and is of broad interest to the community

---

### Official Review · Reviewer_GkKj · 2023-11-22

**Novelty:** 6
**Technical Quality:** 3

**Review:**

This paper studies the imbalanced graph classification problem. Specifically, it aims to address both the class imbalance and size imbalance problem in one method, based on the adaptive data augmentation and Graph of Graphs (GoG) propagation.

Quality: Fair

Clarity: Most of the presentation is clear but the Section 3.3 is text-heavy and more mathematical presentations are preferred.

Originality: Good

Significance: Good. It is an important problem and the solution is novel.

**Pros**:

P1. The proposed solution is novel.

P2. The presentation is good, except Section 3.3 as I previously mentioned.

P3. The experiments show good performance.

**Cons**

C1. The proposed method to propagate embeddings between GoG as a data augmentation module for graph-level tasks is interesting. However, I have several questions for authors about this part.

C1 (1) As Eq. (12) shows, the graph embeddings are propagated on the (higher-level) GoG, how to ensure the graph embeddings contain all the needed graph information (e.g., size, average degree)? This question is related to the expressiveness of the graph encoding GNNs.

C1 (2) Assume there are only two graphs (in the constructed GoG), a large-size graph and a small-size graph. Then, after the message-passing on the GoG as Eq. (12) shows, what is the meaning of the smoothed (diffused) graph embeddings? Do they represent "middle-size" graphs? If so, do you think such a "smoothing of the graph size" can benefit downstream tasks? My intuitive understanding is that the test performance will degrade because based on the regular assumption, the test graphs have the same distribution as the training graphs (e.g., a large-size test graph and a small-size test graph), but the smoothed training graph embeddings present two "middle-size" graphs. This problem is related to out-of-distribution generalization.

C1 (3) Following the previous question, looks like the proposed GoG construction can only work for transductive training. If the test graphs are not presented during the training, how to propagate the test graphs' embedding on the constructed GoG.

I can consider improving my evaluation of the "Technical Quality" if the above questions are well-answered.

C2. As I mentioned, the construction of the Graph of Graphs (GoG) Section 3.3 seems to be the core of the proposed method, which should be better presented. Now it is text-heavy and requires the audiences to be very familiar with methods such as shortest path kernel and CNARW, which lowers the readability of this section.

C3. This is a minor con: the proposed augmentation methods in Section 3.2 are purely heuristic-based

**Questions:**

Please see my review part.

**Reviewer Confidence:**

4: The reviewer is certain that the evaluation is correct and very familiar with the relevant literature

**Scope:**

4: The work is relevant to the Web and to the track, and is of broad interest to the community

---

### Official Review · Reviewer_AqqQ · 2023-11-23

**Novelty:** 7
**Technical Quality:** 7

**Review:**

This paper analyzes the class imbalance and structure imbalance problems that exist in graph classification, which is a very novel perspective. In the analysis of structure imbalance, different modules are also used to solve the problem of graph average degree imbalance and size imbalance and have good performance on different benchmarks.

Strengths
S1.  The motivation proposed in this paper is to look at the imbalance phenomenon in graph classification from different perspectives. In the discussion of imbalanced problems on graphs, more attention is now paid to the node classification imbalance, such as node class imbalance, node degree imbalance, etc., but there is little discussion of graph classification imbalance. This paper considers the problem of graph class imbalance and graph structure imbalance at the same time, which is inspiring for future work.
S2.  The DoOA and SoGoG are highly original and intuitively effective which can solve the structrue imbalance phenomenon. The training process of DoOA is well clarified.
S3.  The paper makes sufficient comparisons with different baselines and proves the effectiveness of each module in solving the imbalance problem through ablation experiments.
S4. This paper is well written and the visualization of the training process allows me to have a more intuitive understanding.

Weaknesses
W1.  The paper lacks some theoretical analysis. Why is it effective to construct graphs of graphs? Will there be noise propagation, which will have a negative impact?
W2.  It would be beneficial if the author could provide a brief explanation or discussion on the scalability of the proposed DoOA, which may face a scenario where the degree distribution is relatively even.

**Questions:**

All my question is mentioned above, I hope the author can give an effective reply.

**Reviewer Confidence:**

4: The reviewer is certain that the evaluation is correct and very familiar with the relevant literature

**Scope:**

4: The work is relevant to the Web and to the track, and is of broad interest to the community

---

### Official Review · Reviewer_wnaB · 2023-11-26

**Novelty:** 4
**Technical Quality:** 4

**Review:**

This work proposes a method called ImbGNN to handle both the class and structural imbalance issues over graph classification. Specifically, it first designs a degree-oriented optimal augmentation to generate graphs with more imbalance degree distribution. Then it proposes a size-oriented GoG to address the class imbalance issue, which allows smaller graphs can have access to more information. Comprehensive experiments over benchmark datasets show the effectiveness of ImbGNN. However, I would like to weakly reject this work due to the following reasons:

Strength:


1. This work is well-written with clear motivations, nice figures, and comprehensive experiments and discussions.

2. The idea of focusing on both class and degree imbalance issues is impressive.




Weakness:

1. The motivation for analyzing the class imbalance issue and degree imbalance issues for the graph classification task is not clearly discussed. Why does this work focus on the class and degree imbalance issues over the graph classification task? Are there any special limitations of these two imbalance issues over the task?

2. Some important baseline methods in graph class imbalance learning should be discussed and compared to show the effectiveness and superiority of the designed model, i.e., GraphSMOTE, GraphSHA, and PC-GNN). Although those works are not specifically designed for graph classification tasks, they can be easily applied to graph classification tasks if necessary, like mean pooling over the node embeddings.

3. Some comparable experiments among ImbGNN and some SOTA in graph degree imbalance learning are required. such as Tail-GNN[1].

4. The format of some equations should be revised and the format of all equations in this work should be unified. There should be a comma or a full stop at the end of each equation. For instance, a full stop should be added to Equation 6, while a comma should be added to Equations 2 and 3.


[1] Tail-gnn: Tail-node graph neural networks, KDD 2021.

**Questions:**

Please refer to the weakness for questions.

**Reviewer Confidence:**

4: The reviewer is certain that the evaluation is correct and very familiar with the relevant literature

**Scope:**

3: The work is somewhat relevant to the Web and to the track, and is of narrow interest to a sub-community

---

### Decision · Program_Chairs · 2024-01-22

**Decision:**

Accept

**Comment:**

This paper proposes a new augmentation approach to simultaneously deal with class imbalance and degree imbalance in GNNs. The results on real datasets demonstrate the superiority of the method, although some reviewers noted that the GoG approach seems a bit adhoc and lacks a clear justification. Overall this is a borderline paper.